# Free-Standing ZnO:Mo Nanorods Exposed to Hydrogen or Oxygen Plasma: Influence on the Intrinsic and Extrinsic Defect States

**DOI:** 10.3390/ma15062261

**Published:** 2022-03-18

**Authors:** Maksym Buryi, Zdeněk Remeš, Vladimir Babin, Sergii Chertopalov, Kateřina Děcká, Filip Dominec, Júlia Mičová, Neda Neykova

**Affiliations:** 1Institute of Physics of the Czech Academy of Sciences, Na Slovance 1999/2, 182 00 Prague, Czech Republic; buryi@fzu.cz (M.B.); babinv@fzu.cz (V.B.); chertopalov@fzu.cz (S.C.); decka@fzu.cz (K.D.); dominecf@fzu.cz (F.D.); 2Department of Nuclear Chemistry, Faculty of Nuclear Sciences and Physical Engineering, Czech Technical University in Prague, Břehová 7, 115 19 Prague, Czech Republic; 3Institute of Chemistry SAS, Dúbravská Cesta 9, 845 38 Bratislava, Slovakia; chemjumi@savba.sk; 4Centre for Advanced Photovoltaics, Faculty for Electrical Engineering, Czech Technical University in Prague, Technická 2, 166 27 Prague, Czech Republic

**Keywords:** zinc oxide nanorods, Mo^5+^, shallow donors, plasma treatment, photo-, radio- and cathodoluminescence, EPR

## Abstract

Cationic doping of ZnO nanorods has gained increased interest as it can lead to the production of materials with improved luminescent properties, electrical conductivity and stability. We report on various Mo-doped ZnO powders of nanorods synthesized by the hydrothermal growth method. Further annealing or/and cold hydrogen or oxygen plasma modification was applied. The atomic structure of the as-grown and plasma-modified rods was characterized by X-ray diffraction. To identify any possible changes in morphology, scanning electron microscopy was used. Paramagnetic point defects were investigated by electron paramagnetic resonance. In particular, two new types of defects were initiated by the plasma treatment. Their appearance was explained, and corresponding mechanisms were proposed. The changes in the luminescence and scintillation properties were characterized by photo- and radioluminescence, respectively. Charge trapping phenomena were studied by thermally stimulated luminescence. Cold plasma treatment influenced the luminescence properties of ZnO:Mo structures. The contact with hydrogen lead to an approximately threefold increase in intensity of the ultraviolet exciton-related band peaking at ~3.24 eV, whereas the red band attributed to zinc vacancies (~1.97 eV) was suppressed compared to the as-grown samples. The exciton- and defect-related emission subsided after the treatment in oxygen plasma.

## 1. Introduction

ZnO nanorods (NRs) are a material with a broad spectrum of uses. In particular, they are implemented in the engineering of solar cells [1,2,3,4], optoelectronics, photonics, gas- [5] and biosensors [6], and photocatalysis [7]. Another important application of the ZnO nanostructures is radiation detectors (scintillators) [8,9]. Moreover, they have the strong potential for ultrafast detection in time-of-flight positron emission tomography [10] due to ultraviolet (UV) exciton emission with the maximum at about 380 nm (3.26 eV) [11,12,13,14,15]. However, a serious drawback of ZnO is the presence of large amount of intrinsic and extrinsic luminescent defects such as differently charged zinc or oxygen vacancies (V_Zn,O_), zinc interstitials or impurities [11,12,16,17]. These luminescent centers either produce radiative emission bands within the about 1–2.4 eV [18,19,20,21] or contribute to non-radiative transitions. Moreover, the detailed analysis of ZnO based nanostructures via electron paramagnetic resonance (EPR) spectroscopy has provided information on the existence of Ga, Al or H shallow donors (SD) producing the typical EPR signal at g ≈ 1.96 [22,23,24]. Importantly, the {Zn^+^ + D} model (D = Ga, Al or H-donor) has been proposed in [20].

Improvement of the exciton emission requires certain material engineering steps, including suppression of the crystal defects [25,26] by modifying conditions of growth [13,14] or choosing proper dopants. The latter could be, for instance, of the donor property serving as effective suppliers of free electrons. Molybdenum with the 3+–6+ row of charge states is an eligible candidate in this case since its improvement of the conductivity of ZnO:Mo nanorods and thin films has previously been observed [20,27,28]. In the past decade post-growth treatment, for example, annealing in air or exposure to hydrogen or oxygen plasma, has appeared the most efficient tool for the moderation of ZnO properties [13,14,19,20,21,29,30,31,32].

The goal of this research is the clarification of luminescence and scintillation properties as well as defect creation processes of hydrogen or oxygen plasma-treated ZnO:Mo nanorods. It should be noted that in Ref. [29], ZnO:Mo(10 and 30%) were studied, whereas presently, the Mo doping level is not higher than 1%. The ZnO:Mo(10%) sample [29] was mostly composed of large microrods—1 μm thick and about 4–5 μm long, while the ZnO:Mo samples in the present case are nanorods about 400–500 nm thick and 1–2 μm long. The ZnO:Mo(30%) sample [29] is not ZnO at all. The ZnO:Mo(0.05, 0.25 and 1%) samples in the present case are from the same batch as reported in Refs. [20,33]. Material phases other than ZnO have been detected as well [20,29,33]. The amount of these phases increased with the Mo doping level [20,29] and therefore, the creation of a Mo-based crystalline substance was expected. However, XRD analysis of the origin of those phases is not provided in Refs. [20,29]; this was only attempted in in Ref. [33]. The material phases were tentatively deduced as follows: H_0.6_MoO_3_, Mo_3_O_8_·xH_2_O and, probably, Zn_2_Mo_3_O_8_ preferentially oriented along {100}. However, the effect of hydrogen and oxygen plasma treatment remained unknown and is addressed only in the current research. This is an important question since plasma treatment induced changes in the material phases present in the ZnO:Er nano- and microrods [21]. XPS analysis was carried out on the ZnO:Mo samples [33] confirming the presence of Mo on the surface of ZnO nanorods as Mo^6+^. The influence of annealing in air at different temperatures in the range from 120 to 700 °C on luminescence and defect creation have been investigated in Refs. [20,29] as well. In particular, the observed red emission band (at about 1.8–2 eV) was significantly suppressed at 120–350 °C and then increased again at 400–500 °C. Further increase in the annealing temperature led to slight decrease in the band [20,29]. This was explained by the two-component origin of the band (ascribed to the neutral zinc vacancy), and the interplay between them and participation of the zinc vacancy in the charge trapping processes [21,34]. Exciton emission at about 3.25 eV was improved by the annealing in air at 350 °C and suppressed at the higher temperatures [20,29]. Moreover, an Mo-related exciton emission peak was detected [20]. Similar effects were observed for ZnO:Mo thin films with similar Mo doping levels [20]. Plasma hydrogenation resulted in an increase in the exciton emission of ZnO:Er nano- and microrods [34]; therefore, it was also the task of the present research, to search for an effective tool for improvement of ultrafast exciton emission. The defect states were also studied by EPR in ZnO:Mo samples as reported in Refs. [20,29] for the samples annealed in air in the 120–700 °C temperature range. The appearance of the Mo^5+^ signal at the isotropic g factor, g = 1.905, from the ZnO host and the increase in its intensity upon annealing in air at 120–350 °C, with a subsequent decrease after annealing at 400–700 °C, indicate the Mo^4+^ to Mo^5+^, and finally to Mo^6+^, charge transformation due to oxidation. The signal intensity increased along with the Mo doping level [20,29]. Moreover, a correlation with the XPS results was observed [33]. X-ray irradiation creates different Mo^5+^ signals originating from the ZnO host with isotropic and axial g tensors, respectively: g(Mo^5+^_1_) = 1.923; g_⊥_(Mo^5+^_2_) = 1.896, g_||_(Mo^5+^_2_) = 1.860 [33]. In the present case, plasma oxidation or hydrogenation resulted in the appearance of different Mo^5+^ signals never reported previously for ZnO:Mo. Moreover, the influence of hydrogen or oxygen plasma treatment on the luminescence properties of the ZnO:Mo host, especially considering the fact that Mo can easily change its charge state, have not been studied yet. The influence of the charge states of the new molybdenum centers on the origin of the 1.8–2 eV red band is not known either. Moreover, since Mo is donor and has influence on exciton emission [20] its effect on the exciton emission under the plasma treatment is of great importance as well. These aspects have never been addressed in previous works and thus they deserve to be considered presently.

## 2. Experimental

### 2.1. Samples Synthesis Procedure

All the chemicals were used as received without additional purification. Zinc nitrate hexahydrate (CAS: 10196-18-6) (Zn(NO_3_)_2_·6H_2_O) and hexamethylenetetramine (CAS: 100-97-0) (HMTA, C_6_H_12_N_4_) were purchased from Slavus (Bratislava, Slovakia)). Ammonium heptamolybdate tetrahydrate was used as the Mo dopant precursor (CAS: 12054-85-2) (NHMO, (NH_4_)_6_Mo_7_O_24_·4H_2_O) from Sigma-Aldrich (Darmstadt, Germany). Deionized water was purified with a system from So~Safe Water Technologies (Dubai, UAE), having a conductivity 0.20 μS cm^−1^ at 25 °C. Mo-doped ZnO nanorods with nominal compositions Zn_1−x_Mo_x_O (x = 0, 0.0005, 0.0025, 0.01) were prepared by the hydrothermal growth method. First, the Zn(NO_3_)_2_·6H_2_O and (NH_4_)_6_Mo_7_O_24_·4H_2_O in the corresponding molar ratio (99.95:0.05 or 99.75:0.25 or 99:1) were dissolved in 250 mL of deionized water, with vigorous stirring by a magnetic stirrer at RT for 30 min. The sample Zn_0.9995_Mo_0.0005_O was prepared from 1.8583 g (24.9876 mM) of Zn(NO_3_)_2_·6H_2_O and 0.0039 g (0.0124 mM) (NH_4_)_6_Mo_7_O_24_·4H_2_O. 1.8545 g (24.9376 mM) of Zn(NO_3_)_2_·6H_2_O and 0.0193 g (0.0624 mM) of (NH_4_)_6_Mo_7_O_24_·4H_2_O was used to prepare sample Zn_0.9975_Mo_0.0025_O. The sample Zn_99_MoO was prepared from 1.8406 g (24.75 mM) of Zn(NO_3_)_2_·6H_2_O and 0.0772 g (0.25 mM) of (NH_4_)_6_Mo_7_O_24_·4H_2_O. The nominal concentration of the Zn_1−x_Mo_x_O (x = 0, 0.0005, 0.0025, 0.01) in the final suspensions was 25 mM. Second, an aqueous solution was prepared of HMTA 0.8762 g (25 mM) (250 mL) (the same amount for the preparation of all samples). All the solutions were filtered through a Whatman 2 filter. Subsequently, the HMTA solution was added to each Zn_1−x_Mo_x_O solution and stirred vigorously at RT for 15 min. Third, the mixture reaction was followed by hydrothermal growth of nanorods by heating at 90 °C for 3 h. The grown nanorods were isolated and then purified (removal of residual salts), by washing 5 times with deionized water and subsequent centrifugation at 11,600 rpm (RCF: 15,500× *g*) for 15 min. Finally, the samples were lyophilized at least 24 h.

### 2.2. Methods of Characterization

The plasma modification was completed in an inductively coupled plasma (ICP) reactor designed for particular research needs in cooperation with SVCS Process Innovation s.r.o. company (Brno, Czechia). The processing parameters were set with times up to 16 min, gas flow of 20 sccm and RF power of 200 W. The electrodeless reactor, operating at radio frequency, 13.56 MHz, is equipped with a tubular quartz chamber placed into a copper coil. Before each plasma treatment, the chamber was evacuated below 0.1 Pa and kept for 15 min, purging with 50 sccm flow of process gas. During the plasma modification, the processed nanopowder was stirred in situ in a grounded stainless-steel cradle-line holder. For better understanding, the following designation of the pristine and plasma modified samples was used: “as-grown” (AG), “hydrogen plasma treated” (PH) and “oxygen plasma treated” (PO).

To define the crystal structure of the nanopowder samples, an X-ray diffractometer (Empyrean, Malvern Panalytical, Malvern, UK,) with Cu K_<α>_ radiation (λ = 1.54151 Å, at U = 45 kV, I = 30 mA) was used. The X-Ray diffraction patterns were determined in the range 2θ = 10–120 °C with a step 0.05° and time per step 5 s and then compared to the relevant records in the ICDD PDF-2 database (version 2013).

EPR measurements were performed with a Bruker EMX plus (9.4 GHz) commercial X-band spectrometer (Bruker, Billerica, MA, USA) with sensitivity about 10^12^ spins/mT. The temperature range was within 4 ÷ 296 K, determined using an Oxford Instruments ESR900 cryostat (Abingdon, Oxfordshire, UK).

Photoluminescence (PL) spectra were recorded in the UV-visible (350–800 nm, 2 nm spectral resolution) detection range. The PL spectra were measured with 5 mg ZnO powder pressed into pellets. A pulsed UV LED (340 nm, 1 mW, 333 Hz) coupled with a narrow band-pass optical filter was used as the source of excitation light. The other components of the PL spectrometer were as follows: spectrally calibrated double gratings monochromator SPEX 1672; long-pass filters (LP350 and LP450); a cooled, red sensitive photomultiplier; a current preamplifier (10 µA/V) and a lock-in amplifier referenced to the LED frequency. The intensity of the back-scattered excitation light dominated at 355 nm was used for the PL spectral intensity normalization.

The radioluminescence (RL) and thermally stimulated luminescence (TSL) measurements were performed in the same conditions, in the spectral range 200–800 nm, using a Horiba Jobin-Yvon 5000 M spectrometer (Horiba Scientific, Kisshoin, Minami-ku, Kyoto, Japan) with an Oxford liquid nitrogen cryostat and a TBX-04 (IBH) photomultiplier. The spectral resolution of the monochromator was 8 nm. The RL and TSL spectra were recorded at 300 K and 77 K, respectively. Irradiation of the sample powders was performed by a Seifert X-ray tube operated at 40 kV with a tungsten target. All the spectra underwent corrections for distortion caused by the experimented setup.

All PL and RL spectra were recalculated from wavelength to photon energy scale by using the Jacobian correction [35].

Cathodoluminescence (CL) images and energy-dispersive X-ray elemental mapping were taken using a Philips XL30 electron microscope (Thermo Fisher Scientific, Waltham, MA, USA).

## 3. Results and Discussion

### 3.1. Phase Purity and Morphology

Analysis of X-ray diffraction patterns of the as-grown ZnO:Mo(0.05, 0.25 and 1%) nanorods (Figure 1A) allowed us to conclude that these samples were not pure phase hexagonal Wurtzite ZnO (record No.00-005-0664) [34], but contain some other, unintentional phases besides the ZnO one.

It can be seen that the intensity of the reflections of these unintentional phases increases with the Mo doping level (Figure 1A). The corresponding patterns were tentatively ascribed to H_0.6_MoO_3_ (record No. PDF01-070-4477), Mo_3_O_8_·xH_2_O (record No. PDF00-021-0574, not present in the ZnO:Mo(0.05%) sample) and, probably, Zn_2_Mo_3_O_8_ phase preferentially oriented along {100} (JCPDS card #01-076-1737, not present in the ZnO:Mo(0.05%) sample) as also reported in Ref. [34]. The XRD patterns are shown in more detail in Appendix A.

The ZnO:Mo(0.05, 0.25 and 1%) samples were exposed to hydrogen and oxygen plasma. The corresponding XRD patterns are plotted in Figure 1B,C. The analysis of these patterns led to the conclusion that the plasma treatments caused no, or negligibly small, changes to the material phases of the ZnO:Mo(0.25 and 1%) samples, i.e., the unintentional phases as well as the ZnO phase underwent almost no changes. In contrast, the H_0.6_MoO_3_, the only non-ZnO phase present in the as grown ZnO:Mo(0.05%) sample, completely vanished in the hydrogen plasma-treated sample and remained unchanged in the oxygen plasma-treated ZnO:Mo(0.05%) sample. A closeup of the corresponding patterns is shown in Appendix A. The disappearance of the H_0.6_MoO_3_ phase after the hydrogen plasma treatment can be explained by the dissociation of the H_0.6_MoO_3_ molecule: H_0.6_MoO_3_ + H_2_ $\overset{\mathrm{PH}}{\to }$ Mo + 1.3H_2_O + 0.85O_2_. Oxygen plasma would have no, or very little, effect of this sort. Therefore, the H_0.6_MoO_3_ phase survived contact with oxygen plasma. Hydrogen plasma treatment also resulted in suppression of the ZnO phase, while plasma oxidation increased the amount of ZnO phase (Appendix A). This can be explained by the presence of surface defects like zinc and oxygen vacancies. This leads to a sheared surface, where dangling bonds are common. Oxygen anions at the surface can miss some of the initial bonding. Therefore, hydrogen ions and radicals from plasma can interact with these oxygen anions, tearing them off the ZnO rods’ surface. Hydrogen ion bombardment resulting in devastation of the ZnO rods surface cannot be fully excluded either [21]. Altogether these effects result in a worsening of the surface and, therefore, the reflections from ZnO phase drop in intensity (see Appendix A). In contrast, oxygen from plasma fills in the oxygen vacancies at the surface of ZnO rods and consequently, the ZnO structure is improved. These effects were pronounced in the ZnO:Mo(0.05%) and negligible in the ZnO:Mo(0.25 and 1%) due to the presence of large amount of Mo covering the surface. This was presently confirmed by EDX (see also XPS results in the previous work [34]). Molybdenum holds more oxygen due to Mo^4+,6+^ oxidation states at the surface and thus acts as a shield from the contact with plasma along with the molybdenum-based material phases. It is known that Mo and Zn easily form complex oxides (see, e.g., Ref. [36]) since Mo possess metallic and non-metallic properties with the ability to form salts. Therefore, the observed segregation of the molybdenum-based material phases occurs.

The treatment of the ZnO nanorods with the oxygen and hydrogen plasmas caused no changes in the morphology of the nanorods, as can be seen in the SEM images in Figure 2 of the ZnO:Mo(0.05%) sample.

It can be seen that the rods preserved their shape (hexagonal prismoids, for more details see original SEM images in Appendix A) and the average size of individual rods (about 400–500 nm thick and over 1 μm long) is approximately the same in the ZnO:Mo(0.05%), as-grown, H_2_ and O_2_ plasma treated samples. The nanorods were grown from a water solution containing dissolved precursors. The solution contained accidental nucleation centers, i.e., “seeds”, very small pieces (maximum tens of nm large) of incompletely dissolved reagents. The solution is kept at such a temperature (maintained during the whole growth process) specific for nucleation to begin. This temperature has been found experimentally [1,2,3,6] after numerous attempts to grow rod-like structures, i.e., the geometry of the rods is the product of the fine balance between the expansion due to growth of the ZnO crystallite and dissolution in the solvent. The corresponding calculations were not carried out.

### 3.2. Paramagnetic States by EPR

EPR spectra measured in the as-grown ZnO:Mo(0.05, 0.25 and 1%) nanorods were composed only of the strong signal at the g factor g = 1.954 (SD_C_ in Figure 3). This has previously been attributed to the shallow donors, {Zn^+^ + D} (D = Ga, Al or H-donor), in ZnO structures [17,18,19,20,22,23,37,38,39]. The SD_C_ signal integral intensity was reduced slightly in the spectra of the ZnO:Mo(0.05%) sample exposed to the hydrogen/oxygen plasma. The effect was a bit more pronounced in the reduction atmosphere (see Figure 3A and Table 1). In contrast, the SD_C_ signal was slightly increased in the ZnO:Mo(0.25%) sample exposed to the hydrogen plasma, whereas treatment with oxygen plasma increased the SD_C_ signal even more (see Figure 3B and Table 1). This trend was much stronger in the ZnO:Mo(1%) sample (see Figure 3C and Table 1). The discussed tendencies are unique to the present set of ZnO:Mo samples.

The shallow donor signal had completely different behavior in the oxygen and hydrogen plasma treated ZnO:Er(0, 0.05, 0.25 and 1%) samples [38]. The SD_C_ EPR signal intensity was, in general, lower in the samples with increased Mo content, as has previously been reported in [20] (see Figure 3 and Table 1).

The contact with hydrogen or oxygen plasma led to the appearance of a new isotropic SD_S_ signal in all the ZnO:Mo(0.05, 0.25, 1%) samples, which was not observed in the as-grown samples (Figure 3). The intensity of the SD_S_ signal was much higher after the interaction with oxygen plasma compared to the hydrogen plasma (Figure 3, Table 1). Moreover, the SD_S_ signal intensity increased along with Mo content (Table 1). To determine the precise g factor value, the SD_S_ signal in the EPR spectrum of the ZnO:Mo(1%) sample exposed to hydrogen plasma was chosen for advanced analysis with the fitting procedure based on the following spin Hamiltonian:(1)$$\hat{H}=\beta \hat{S}\hat{g}H,$$
where the letters in the right part of the expression stand for Bohr magneton, electron spin operator (electron spin $\hat{S}$ = 1/2), **ĝ** tensor and magnetic field vector, respectively. The isotropic **ĝ** tensor was used for the SD_S_ signal simulation. The g factor, g = 2.0036 ± 0.0003, was thus determined from the fit. Both experimental and fitting spectra are plotted in Figure 4. Notably, EPR signals with g = 2.0038 [13] and g = 2.0048 [40] have previously been detected in undoped ZnO. Moreover, the EPR signal at g = 2.0038 was also measured in the room-temperature X-ray-irradiated ZnO:Mo(0.05, 0.25 and 1%) nanorods [20]. Its intensity was similarly dependent on the Mo doping level [20] as the SD_S_ was in the present case (Table 1).

All of this allowed us to attribute the SD_S_ signal in Figure 3 to the shallow donors from the nanorods’ shell. The SD_C_ signal originates from the shallow donors in the nanorods’ core. By using the spin-Hamiltonian in Equation (1), the experimental SD_C_ signal has been approximated with the fitting one. The precise g factor was thus determined to be g = 1.954 ± 0.001. The fit is shown in Figure 4.

Interestingly, plasma treatment also resulted in the Mo^5+^ signal appearance at g~1.9, as is common for Mo^5+^ (4d^1^ outer shell) [39] (for more details see [20,29]), in the ZnO:Mo(0.25 and 1%) samples. This signal increased along with the Mo doping level. The hydrogen plasma had the strongest influence on it while the oxygen plasma had no (ZnO:Mo(0.25%)), or very little (ZnO:Mo(1%)) effect, as can be seen in Figure 3. The two main features of the Mo^5+^ signal are indicated by vertical dashed line segments in the corresponding insets in Figure 3B,C. Based on these observations it is evident that Mo influences the charge transfer processes invoked by plasma treatment. The Mo^5+^ EPR signal is very broad, covering the 250 G magnetic field range (see e.g., Figure 3C). Therefore, in order to find out the spin Hamiltonian parameters, the fitting procedure must be applied (Equation (1)). Logically, the EPR spectrum of the ZnO:Mo(1%) sample exposed to the hydrogen plasma and composed of the SD_C,S_ and Mo^5+^ signals should be used for this purpose. Fitting of these three signals simultaneously using Equation (1) (the fit of the SD_C,S_ has already been discussed and shown above in Figure 4) allowed more accurate determination of the spin Hamiltonian parameters (Equation (1)). As one can see in Figure 4, the fit is fairly good. The Mo^5+^ signal (Figure 3B,C) turned out to be two overlapping spectra, ${\mathrm{Mo}}_{1}^{5+}$ and ${\mathrm{Mo}}_{2}^{5+}$, as shown in Figure 4. The fit parameters (**g** tensor components) were: g_a_ = 1.945 ± 0.001, g_b_ = 1.926 ± 0.001, g_c_ = 1.888 ± 0.001 (${\mathrm{Mo}}_{1}^{5+}$) and g_a_ = 1.925 ± 0.001, g_b_ = 1.913 ± 0.001, g_c_ = 1.905 ± 0.001 (${\mathrm{Mo}}_{2}^{5+}$). Hyperfine structure due to the ^95,97^Mo nuclei was not resolved, and therefore it was not considered (for more details see [20]).

The observed trends exhibited by the SD_C,S_ and Mo^5+^ signals can be explained by the following models. According to [20,22,23,24], the origin of the SD signal is Zn^+^ + D, D = H, Ga or Al. The decrease in the SD_C_ signal after contact with hydrogen or oxygen plasma in the ZnO:Mo(0.05%) may thus be related to the creation of Zn^2+^ or Zn^0^ (metal zinc with the outer shell 2s^2^) from Zn^+^ in the following two ways:(2)$${\mathrm{Zn}}^{+}+D+e^{-}\overset{\mathrm{PH}}{\to }{\mathrm{Zn}}^{0}+D$$
(3)$${\mathrm{Zn}}^{+}+D+h^{+}\overset{\mathrm{PO}}{\to }{\mathrm{Zn}}^{2+}+D$$

Both Zn^0^ and Zn^2+^ are not paramagnetic, i.e., EPR silent.

The increase in the SD_C_ signal and creation of the Mo^5+^ due to the influence of hydrogen or oxygen plasma on the ZnO:Mo(0.25, 1%) samples may be described by the following two mechanisms:(4)$${\mathrm{Zn}}^{2+}+D+{\mathrm{Mo}}^{6+}+2e^{-}\overset{\mathrm{PH}}{\to }{\mathrm{Zn}}^{+}+D+{\mathrm{Mo}}^{5+}$$
(5)$${\mathrm{Zn}}^{2+}+{D+2\mathrm{Mo}}^{4+}+2h^{+}\overset{\mathrm{PO}}{\to }{\mathrm{Zn}}^{+}+{D+\mathrm{Mo}}^{6+}+{\mathrm{Mo}}^{5+}$$

The creation of the Mo^5+^ is less probable in (5) due to the weak Mo^5+^ EPR signal of the oxygen plasma treated ZnO:Mo(1%) and its total absence in the rest of the samples (Figure 3). The existence of Mo^4+^ in the as-grown ZnO nanorods was confirmed in [34]. Mechanisms (4) and (5) can also explain the creation and increase in the SD_S_ signal with increased Mo doping level (Figure 3) after both types of plasma treatment (the larger the molybdenum content, the more Mo^4+^ is expected according to [20]). Mechanism (5) must play a more significant role in SD_S_ signal creation since it is stronger in the oxygen plasma-treated samples (Figure 3). It is noteworthy that the plasma treatments caused a decrease in the SD_C_ and an increase in the SD_S_ signal intensities in the ZnO:Mo(0.05%) sample compared to the ZnO:Mo(0.25 and 1%) samples. Therefore, one may expect a charge exchange between the core (SD_C_) and shell (SD_S_) ZnO nanorod regions in this case. However, it is less probable in the other two samples with 0.25 and 1% of Mo due to the larger Mo contents resulting in an increased charge trapping radius, leading to the creation of Mo^5+^ (Figure 3B,C).

### 3.3. Photoluminescence Properties

Plasma hydrogenation or oxidation influence the luminescence and defect creation processes in ZnO [15,18,20,38,41], as can be observed in the example of the PL spectra measured in ZnO:Mo(0.05, 0.25, 1%) nanorods prior to and after the exposure to hydrogen or oxygen plasma (Figure 5).

Two emission bands are clearly visible: E_red_ (2.00 eV) and E_exc_ (3.25 eV). The E_exc_ band was ascribed to the exciton emission [11,12,13,14,15]. It was strongest in the as-grown ZnO:Mo(0.25%) sample (by a factor of two compared to the as-grown ZnO:Mo(0.05 and 1%) samples) as has previously been reported in Ref. [20]. The E_exc_ intensity ratio in the as-grown ZnO:Mo(0.05, 0.25 and 1%) is 1:2:1. Mo acts as a donor in ZnO [20,27,28]. Its 4d levels appear below the bottom of the conduction band mixing with the exciton states [20]. Note, that the Mo-bound exciton existence was considered in Ref. [20] as well. There, the intensity of this exciton PL peak decreased with increasing Mo doping levels [20]. Therefore, the rise in the exciton emission and its drop upon the increased Mo doping level from 0.05% through 0.25% to 1% can be explained by the strong interference between the Mo donor levels and the exciton states. At low Mo doping levels, the more Mo donor levels supply electrons to the exciton states aiding the exciton emission (at 0.05 and 0.25%). At 1% of Mo, more of the Mo appears in the form of different material phases on the surface of the ZnO rods and the dense Mo states start pumping out electrons from the exciton levels to the Mo-based emission centers in the Mo-based material phases at the ZnO-Mo-based phase boundary leading to non-radiative transitions. As a result, a drop in the exciton emission can be observed (Figure 5).

The hydrogen plasma treatment strongly influenced the intensities of the exciton bands in the samples. The exciton band increased by about 3.5 times in the ZnO:Mo(0.05%), by about 1.5 times in the ZnO:Mo(0.25%) and by about 2 times in the ZnO:Mo(1%) samples after the exposure to hydrogen plasma. The E_exc_ intensity ratio in the ZnO:Mo(0.05, 0.25 and 1%) became 3:3:2. The strongly improved exciton emission in the ZnO:Mo(0.05%) correlates with the removal of the H_0.6_MoO_3_ phase (see Appendix A). The improvement in the E_exc_ in the ZnO:Mo(0.25 and 1%) samples can be related to the Mo^5+^ creation by the reduction from Mo^6+^ to Mo^5+^ (Equation (4)). The more Mo^5+^ the stronger the increase in the exciton emission. The paths for charge transfer between the ZnO-Mo-based material phase boundaries discussed above are thus partially removed. There are two Mo^5+^ centers (${\mathrm{Mo}}_{1,2}^{5+}$, see EPR results above) that differ from the ones observed after the annealing in air or X-ray irradiation of the ZnO:Mo nano- and microrods [20]. Based on all these considerations, the ${\mathrm{Mo}}_{1,2}^{5+}$ centers are expected to originate from the ZnO-Mo-based material phase boundaries.

Oxygen plasma treatment had the opposite effect—the exciton emission intensity dropped by about 2 times in the ZnO:Mo(0.05 and 1%) and by about 4 times in the ZnO:Mo(0.25%). The E_exc_ intensity ratio in the ZnO:Mo(0.05, 0.25 and 1%) became 1:1:1. The observed peculiarities point to the existence of a “depot” for the excitons, regulated by the presence of Mo. Oxygen from the plasma supplies the holes according to Equations (3) and (5) which, by trapping at unknown acceptor levels in ZnO, will deactivate donor-acceptor pairs and as a result the exciton emission is suppressed.

The E_red_ band intensity ratio in the as-grown ZnO:Mo(0.05, 0.25 and 1%) was 1:2:1 (Figure 5) as has also been reported in the previous work [20]. The E_red_ band was similarly suppressed after contact with the hydrogen or oxygen plasma in all the samples, compared to the as-grown ones (Figure 5). A threefold reduction in the E_red_ band intensity was observed for the ZnO:Mo(0.25%) sample (Figure 5B) whereas a less pronounced reduction (around twofold) was observed in the ZnO:Mo(0.05%) and ZnO:Mo(1%) samples (Figure 5A,C). The E_red_ band intensity ratio in the ZnO:Mo(0.05, 0.25 and 1%) after the plasma treatment became 4:4:3. Therefore, the number of V_Zn_^0^ (the origin of E_red_ [21]) must have been maintained at about the same level in the mentioned samples. The E_red_ band is composed of two components as discussed in [21,38]. To determine these components and the distribution of energy between them as a function of the Mo content, the Gaussian expression was used:(6)$$C_{1,2}(E)=G_{1,2}\exp\left(-4\ln2\frac{{\left(E-E_{\max1,2}\right)}^{2}}{FWHM_{1,2}^{2}}\right)$$where the $E_{\max1,2}$ are the corresponding bands maxima, $FWHM_{1,2}$ is the full width at half maxima and $G_{1,2}$ are amplitudes. The determined parameters are listed in Table 2.

To demonstrate the $C_{1,2}(E)$ components and the accurancy of the fit, the decomposition of the PL spectra of the ZnO:Mo(0.25%) sample, as-grown and exposed to hydrogen or oxygen plasma, are shown in Figure 6 along with the calculated peaks.

The maxima positions of the $C_{1,2}(E)$ bands ($E_{\max1,2}$) remain unchanged and located at 1.84 and 2.10 eV, respectively, no matter the type of applied treatment (Table 1). Similarly, the FWHM of the $C_{1,2}(E)$ bands underwent no changes. The intensity ratio of the $C_{1,2}(E)$ bands (I($C_{1}(E)$)/I($C_{2}(E)$)) changed with the treatment used (see Table 1). For the ZnO:Mo(0.05, 0.25 and 1%) samples (as-grown and oxygen plasma-treated) as well as ZnO:Mo(0.25 and 1%) samples exposed to hydrogen plasma, this averaged ratio is approximately the same~2.5. In the H_2_ plasma-treated ZnO:Mo(0.05%) sample, the ratio is 2. The $C_{1,2}(E)$ bands have the following origins [18,19,21]: $C_{1}(E)$ − V_Zn_^0^; $C_{2}(E)$ − (V_Zn_ + D)^0^, where D is some defect. Both bands were weaker after all the treatments mentioned (Figure 5) and zinc vacancies became partly deactivated. The increase in the $C_{2}(E)$ band compared to the $C_{1}(E)$ one in the hydrogen or oxygen plasma-treated samples should be a result of the presence of Mo charge states in the ZnO host. The Mo charge states discussed above in Equations (4) and (5) should participate in the charge compensation thus changing the total number of zinc vacancies, i.e., the incorporation of Mo must play some role in the stabilization of the V_Zn_ and its charge state [20,21]. ZnO, in general, must contain plenty of differently charged zinc vacancies, V_Zn_^2−^, V_Zn_^−^, V_Zn_^0^, V_Zn_^+^, V_Zn_^2+^, which might have produced luminescence as reported in Ref. [19]. The initial charge state of molybdenum, i.e., Mo^4+,6+^ at zinc sites [20,34] and extra positive ${\mathrm{Mo}}_{\mathrm{Zn}}^{2+,4+}$, respectively, would require charge compensation in the first row by the positively charged zinc vacancies: (i) ${\mathrm{Mo}}_{\mathrm{Zn}}^{2+}$ = V_Zn_^2+^ + 2e^−^ = V_Zn_^0^, ${\mathrm{Mo}}_{\mathrm{Zn}}^{2+}$ = 2V_Zn_^+^ + 2e^−^ = 2V_Zn_^0^, 2${\mathrm{Mo}}_{\mathrm{Zn}}^{2+}$ = V_Zn_^2+^ + V_Zn_^+^ + 3e^−^ = 2V_Zn_^0^; (ii) ${\mathrm{Mo}}_{\mathrm{Zn}}^{4+}$ = 2V_Zn_^2+^ + 4e^−^ = V_Zn_^0^, ${\mathrm{Mo}}_{\mathrm{Zn}}^{4+}$ = 4V_Zn_^+^ + 4e^−^ = 4V_Zn_^0^, ${\mathrm{Mo}}_{\mathrm{Zn}}^{4+}$ = V_Zn_^2+^ + 2V_Zn_^+^ + 4e^−^ = V_Zn_^0^. These mechanisms are likely responsible for the increase in the E_red_ PL [21]. Conversely, the following mechanisms can reduce the E_red_ band [21]: (iii) ${\mathrm{Mo}}_{\mathrm{Zn}}^{2+}$ = V_Zn_^0^ + 2e^−^ = V_Zn_^2−^, ${\mathrm{Mo}}_{\mathrm{Zn}}^{2+}$ = V_Zn_^−^ + e^−^ = V_Zn_^2−^, ${\mathrm{Mo}}_{\mathrm{Zn}}^{2+}$ = V_Zn_^0^ + V_Zn_^−^ + e^−^ = 2V_Zn_^−^; (iv) ${\mathrm{Mo}}_{\mathrm{Zn}}^{4+}$ = 2V_Zn_^0^ + 4e^−^ = 2V_Zn_^2−^, ${\mathrm{Mo}}_{\mathrm{Zn}}^{4+}$ = 4V_Zn_^0^ + 4e^−^ = 4V_Zn_^−^, ${\mathrm{Mo}}_{\mathrm{Zn}}^{4+}$ = V_Zn_^0^ + V_Zn_^−^ + 3e^−^ = V_Zn_^2−^ + 2V_Zn_^−^, etc. Mechanisms (i–iv) govern the E_red_ PL intensity observed in the as grown samples, i.e., when mechanisms (i) and (ii) dominate over (iii) and (iv), E_red_ increases, as was observed in the as-grown ZnO:Mo(0.25%) samples in comparison with the as-grown ZnO:Mo(0.05 and 1%) samples. The larger amount of Mo^6+^ in the as-grown ZnO:Mo(1%) compared to the as-grown ZnO:Mo(0.05, 0.25%) [34] will have the consequence that mechanisms (iii) and (iv) dominate over the mechanisms (i) and (ii).

Plasma hydrogenation by supplying extra electrons to V_Zn_^−^, V_Zn_^0^, V_Zn_^+^ and V_Zn_^2+^ leads mostly to reduction of these zinc vacancies to V_Zn_^−,2−^, as E_red_ was strongly suppressed (Figure 5). Moreover, for further charge compensation, the charge state of the molybdenum becomes Mo^5+^ (at least at the ZnO-Mo-based phase boundaries) and Zn^2+^ becomes Zn^+^ (by Equation (4); this also correlates well to the increased shallow donor SD_S_ EPR signal), as discussed above. As a consequence, the mechanisms (i) and (ii) are suppressed and (iii) and (iv) become dominant.

Plasma oxidation supplies oxygen to fill the oxygen vacancies and the holes to be trapped at V_Zn_^2−^, V_Zn_^−^, V_Zn_^0^ and V_Zn_^+^. New oxygen ions at the initially existing oxygen vacancies (already compensated by some other mechanism, i.e., zinc vacancy) will create extra negative charge in the balanced system, at least at the surface. For better charge compensation, the dominant oxidation of the zinc vacancies to V_Zn_^+,2+^ is expected, as the E_red_ was strongly suppressed (Figure 5). For further charge compensation (less pronounced than in case of the hydrogen plasma treatment), the charge state of the molybdenum is becomes Mo^5+^ (at least at the ZnO-Mo-based phase boundaries, Equation (5)), as discussed above. As a consequence, mechanisms (i–iv) are suppressed and the Zn^+^ is created to compensate for Mo^4,5,6+^ (by Equation (5); this also correlates well with the increased shallow donor SD_S_ EPR signal).

The effects of the hydrogen plasma treatment are, in general, approximately the same in the ZnO nanorods deposited on a substrate with a nucleation layer, as presently observed in the free-standing structures (see also Refs. [13,34]), i.e., the exciton-related emission band increases while the defect-related band becomes suppressed several times [42,43]. However, the influence of oxygen plasma is a bit different in the deposited ZnO nanorods as compared to the free-standing ones (see Figure 5 and Ref. [34]). Short exposure time (30 s) resulted in the exciton emission growing to several times its previous intensity, and the defect-related emission being quenched. In contrast, longer treatment (over 30 s up to 600 s) led to a less pronounced effect, and even a decrease in the exciton band and an increase in the defect-related one were observed [44]. Presently, and as reported in Ref. [34], contact between free-standing ZnO nanorods and oxygen plasma always resulted in a strong decrease in both exciton- and defect-related emission bands no matter how short the exposure time was. Based on these observations one may conclude that the deposited arrays of ZnO nanorods must have a better surface with fewer defects due to the dense arrangement (the rods in the inner parts of the deposited arrays are protected from damage by the outer rows) than the free-standing ones.

### 3.4. Radioluminescence

The influence of the contact with the hydrogen or oxygen plasma on the energy conversion and transfer processes was studied in the ZnO:Mo nanorods by measuring radioluminescence. The corresponding spectra are plotted in Figure 7.

The RL spectrum is broad and complex for the as-grown ZnO:Mo(0.05, 0.25, 1%) samples, which were not exposed to any treatment. The clearly visible maximum is observed at approximately 2.58 eV (P band in Figure 7). The P band, which is superimposed on the E_red_ band (similar to PL spectra in Figure 5), may originate from phases other than the ZnO one resolved by XRD and discussed in the Section 3.1 (see Figure 1), especially taking into account the increase in its intensity with respect to Mo content (Figure 7). The P band disappears completely in the RL spectra of the plasma-treated samples, whereas the phases discovered by XRD remain. Therefore, one may conclude that the emission center responsible for the P band was passivated by the plasma.

The E_red_ band measured in the hydrogen or oxygen plasma-treated ZnO:Mo(0.05%) increased about twofold and fourfold, respectively, compared to the as-grown sample (Figure 7A). Hydrogen plasma had almost no effect on the E_red_ band of the ZnO:Mo(0.25%), while it increased about fourfold after the oxygen plasma treatment (Figure 7B). The E_red_ band was increased a small amount in the RL spectrum of the ZnO:Mo(1%) after the oxygen plasma treatment (Figure 7C). In contrast, the E_red1_ band (~1.77 eV) was detected instead of the E_red_ band in the RL spectrum of the hydrogen plasma-treated ZnO:Mo(1%) sample (Figure 7C). It is slightly weaker than the E_red_ band. These newly observed dependences of the defect emission as a function of the Mo doping content strongly differ from those detected in the ZnO:Er(0, 0.05, 0.25 and 1%) reported in [38]. This is in good agreement with the considerations of the two-component origin of the red bands and the interplay between their components’ intensity involving Mo, as discussed in the subsection above for the photoluminescence. The E_red1_ band is expected to originate from the V_Zn_^0^ somewhat perturbed, most probably, by Mo.

The E_red_ band in PL and RL spectra demonstrate different trends upon the plasma treatment (see Figure 5 and Figure 7). This disproportion of the effects in PL and RL has previously been reported and explained for ZnO:Er(0, 0.05, 0.25 and 1%) in [38] by the creation of larger amount of V_Zn_^0^ and participation of the interstitial Zn_i_. This can be confirmed by the experimental observations showing that the PL of ZnO:Er(0.05, 0.25 and 1%) or ZnO:Mo(0.05, 0.25, 1) is sensitive to external X-ray irradiation [34,45]. Molybdenum is known to participate in charge trapping processes as well [45].

The E_exc_ could not be resolved in the RL spectra of the as-grown samples (Figure 7), however, although very weak, it becomes clearly visible after the hydrogen or oxygen plasma treatment in the ZnO:Mo(0.05, 0.25%). This correlates very well with the tendencies observed for the PL spectra in Figure 5. Surprisingly, the RL spectrum of the ZnO:Mo(1%) contains very a weak exciton emission band only after contact with the oxygen plasma while a new blue band, peaking at about 2.78 eV (E_b_ in Figure 7C), appeared in the ZnO:Mo(1%) only after the exposure to the hydrogen plasma. Note that a similar blue band was detected in the RL spectra of ZnO:Er(1%) microrods [38]. It was ascribed there to zinc interstitials, producing blue emission at 2.76 eV [39]. The origin of the E_b_ observed in this work is expected to be similar.

PL and RL underwent the strongest changes after the ZnO:Mo samples’ interaction with hydrogen plasma (see Figure 5 and Figure 7C, the enhancement of exciton emission and the blue band (E_b_) and the E_red1_ band appearance). Therefore, to find out the peculiarities of the RL spectra upon cooling, the hydrogen plasma-treated ZnO:Mo(0.05, 0.25, 1%) samples were chosen (see Figure 8).

It was found that the position of the E_red_ band varied from 2 eV to 1.85 eV when cooling down the ZnO:Mo(0.05 and 0.25%) samples from 295 K to 77 K (Figure 8A,B). This effect can be explained by the fact that the bandgap of the ZnO:Mo(0.05 and 0.25%) samples is wider due to the smaller number of Mo states contributing to the bandgap, whereas in the ZnO:Mo(1%), the donor states created by Mo are more dense, therefore the bottom of the conduction band must be lower. Note, that the excited state of the zinc vacancy-based defects are expected to appear at about 0.5 eV above the bottom of the conduction band [20] in the undoped ZnO; at the same time the Mo-based electron traps were found at about 0.25 eV below the conduction band in the ZnO:Mo rods [45]. Moreover, there are Mo charge traps which are stable at room temperature, i.e., they should be found far below the conduction band. Moreover, there are plenty of oxygen-based acceptor-like defects in the ZnO:Mo rods [45]. The number of these states increased with Mo content [45]. These are the reasons for the E_red1_ band’s appearance in the RL of the ZnO:Mo(1%) sample (Figure 7C and Figure 8C). The variation in the bandgap in the ZnO:Mo(1%) is thus expected to have a less pronounced effect on the ground and excited states of the V_Zn_^0^ in RL with the temperature decrease in comparison with the ZnO:Mo(0.05 and 0.25%). As was expected, the intensity of the E_red_ band (ZnO:Mo(0.05, 0.25%), Figure 7A,B) and E_red1_ band (ZnO:Mo(1%), Figure 7C) increased upon cooling.

The E_exc_ band was exceedingly weak in the ZnO:Mo(0.05%) sample, and strongest in the ZnO:Mo(0.25%) sample, as can be seen in Figure 8A,B. Low temperature led to improvement of the E_exc_ band in the ZnO:Mo(0.25, 1%) samples (Figure 7 and Figure 8). Moreover, the E_exc_ band shifted upon cooling (295–77 K) from 3.18 eV to 3.31 eV, as reported in [20].

The E_b_ band was measured only in the ZnO:Mo(1%) sample at 295 and 275 K (Figure 8C). No exciton band was detected in the RL spectra apart from the E_b_ band (see the inset in Figure 8C) at these temperatures. However, at 250 K and below, the E_exc_ band is present, while the E_b_ band is not. This indicates the possibility of charge and/or energy transfer between the excitons and the Zn_i_ as the origin of the E_b_ band. Interstitial zinc creates shallow donor levels, and the emission occurs between them and the valence band [38,39]. Therefore, reabsorption of the exciton luminescence by the Zn_i_ states is possible (see also the mechanism for the increased red band described in [38]). With the decrease in temperature, the bandgap widens, therefore, more exciton levels contribute to the luminescence mechanisms. The density of the exciton states must overwhelm the Zn_i_ states and thus no electrons are supplied to Zn_i_. As a result, the exciton emission becomes visible while the blue emission is quenched. In contrast, the PL spectra do not contain the blue band. This can be explained by the number of Zn_i_ below the detection limit, as also mentioned in [38].

### 3.5. Charge Trapping States Revealed by TSL

By analogy with the RL above, the charge trapping processes in the hydrogen plasma-treated ZnO:Mo(0.05, 0.25, 1%) samples were studied by TSL (see the corresponding glow curves in Figure 9).

The TSL glow curves are represented by one strong peak due to several contributions which have been reported to be oxygen-based electron or hole trapping centers [33]. This TSL peak was shifted to higher temperatures upon hydrogen plasma treatment, compared to the as-grown samples: (i) from 112 to 121 K in the ZnO:Mo(0.05%); (ii) from 107 to 112 K in the ZnO:Mo(0.25%) sample and (iii) from 106 to 116 K in the ZnO:Mo(1%) sample. The hydrogen plasma treatment strongly improved the intensity of the peak (by about one order of magnitude) in all cases. This is in strong contrast with the TSL glow curves of the ZnO:Er(0.05 and 0.25%) samples where the TSL peaks were decreased after exposure to hydrogen plasma [38]. However, the effects observed in the TSL curves correlate very well with the ZnO:Er(1%) [38]. The increased number of trapping states was to be expected in the present case since the intensity of the E_red_ and E_red1_ bands (zinc vacancies serve as recombination centers for the charge carriers released from traps [34,38,45]) exhibit different trends than the TSL peaks.

In addition, a very small “shoulder-like” feature was observed in the 170–210 K region in the glow curve of the ZnO:Mo(1%) sample (Figure 9C). This feature has been studied in detail in [33].

### 3.6. Spatial Variation of Composition and Cathodoluminescence

The CL images, taken at room temperature and excited by a focused scanning electron beam, show notable contrast in brightness. This correlates with elemental composition (EDX), which was most pronounced in the sample with high Mo content, i.e., ZnO:Mo(1%), as compared in the panels of Figure 10, which were taken from a single place.

The CL image roughly outlines the facets of ZnO crystals (Figure 10A,B), but no obvious relationship between any crystal feature and CL intensity can be drawn. Partial spatial correlation between the SE and CL images (Figure 10A,B) and Mo EDX signal (Figure 10C) can be observed. This suggests that the molybdenum preferably stays in relatively small rods. Comparing the EDX signals for molybdenum and zinc (Figure 10C,D), one may conclude that the Mo-rich rods also contain zinc. Therefore, the incorporation of Mo into the small ZnO rods can be expected. In contrast, the larger ZnO crystals seem not to contain molybdenum in a concentration detectable by EDX, and the weak EDX signal for Mo measured from their surface can be attributed to background noise. The Mo concentration was roughly estimated to be 0.04 ± 0.01 at.% in ZnO:Mo(0.05%), 0.24 ± 0.05 at.% in ZnO:Mo(0.25%) and 0.8 ± 0.2 at.% in the ZnO:Mo(1%) sample.

## 4. Conclusions

Free-standing, hydrothermally grown ZnO:Mo(0.05, 0.25 and 1%) nanorods in powder form were synthesized. Their further modification by annealing in air or/and cold hydrogen or oxygen plasma did not alter their morphology and phase composition, as confirmed by XRD results. The hydrogen plasma caused an increase in the E_exc_ emission band (peaking at ~3.24 eV) up to three times in treated samples as compared to the as-grown nanorods, as shown by PL. The opposite effect was observed after contact with oxygen plasma, where the exciton emission dropped by about three times. The E_red_ band appeared suppressed by the contact with cold plasma as the strongest suppression (by about three times) was observed in the ZnO:Mo(0.25%).

In the samples with the highest amount of Mo, the exciton band was observed only after plasma oxidation, whereas the E_b_ band originating from Zn_i_ (peaking at ~2.78 eV) was detected after the hydrogen plasma treatment at 295 and 275 K. With a decrease in temperature down to 250 K, E_b_ was replaced by E_exc_, which indicates the possibility of charge and/or energy transfer between the excitons and the origin of the E_b_ band. The TSL peak and E_red_ RL intensities were in good correlation; therefore, one may conclude that the interaction with hydrogen plasma lead to improved charge trapping properties in the ZnO:Mo nanorods.

The hydrogen plasma treatment resulted in a decrease in the typical EPR shallow donor signal, whereas it became stronger after contact with oxygen plasma. Remarkably, after the plasma treatment, a new SD_S_ signal appeared (at g ≈ 2.0038), as a counterpart to the SD_C_ signal. In accordance with the core–shell model, the SD_C_ signal was attributed to the ZnO rods’ bulk and the SD_S_ signal to the shell, i.e., the surface of the rods.

The Mo^5+^ signal (at g~1.9) appeared to be due to the trapping of an electron at Mo^6+^ after plasma treatment of the 0.25% doped samples, and its intensity increased with the increase in Mo doping to 1%. The hole trapping at Mo^4+^ and Zn^+^ should originate from the V_Zn_^0^, i.e., the V_Zn_^0^ ↔ Mo^4+,6+^ charge transfer mechanism takes place. Moreover, the larger amount of Mo in the samples creates shielding effect over the ZnO rods’ surface; therefore, the Mo predominately undergoes modification in the contact with plasma.

## Figures and Tables

**Figure 1 materials-15-02261-f001:**
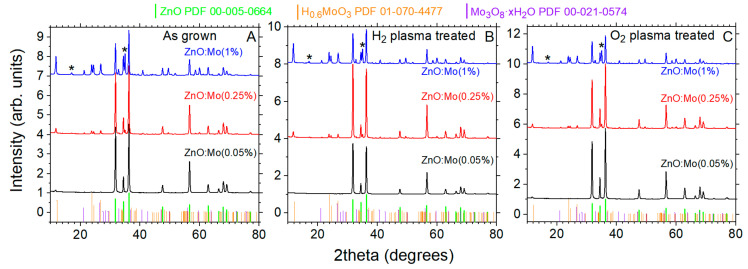
The XRD patterns of the ZnO:Mo(0.05, 0.25 and 1%) rods as grown (**A**), and H_2_ (**B**) and O_2_ (**C**) plasma-treated samples. The reflections of the hexagonal Wurtzite ZnO, H_0.6_MoO_3_ and Mo_3_O_8_ xH_2_O phases are additionally indicated. Asterisks (*) indicate possible Zn_2_Mo_3_O_8_ phase oriented along {100} [33].

**Figure 2 materials-15-02261-f002:**
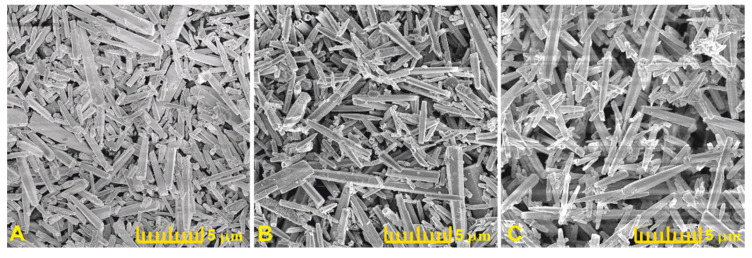
SEM images of the ZnO:Mo(0.05%) powder samples: (**A**)—AG; (**B**)—PH; (**C**)—PO (see Experimental).

**Figure 3 materials-15-02261-f003:**
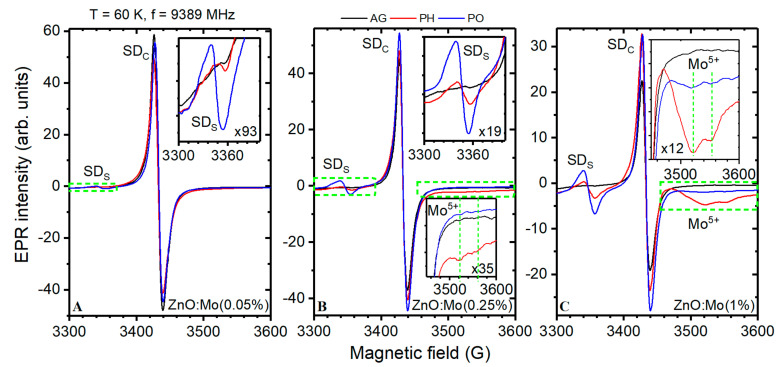
EPR spectra measured at 60 K in the ZnO:Mo(0.05%) (**A**), ZnO:Mo(0.25%) (**B**), ZnO:Mo(1%) (**C**) AG, PH or PO samples. Two dashed vertical line segments indicate the observed specific Mo^5+^ signal components in the insets of panels B and C.

**Figure 4 materials-15-02261-f004:**
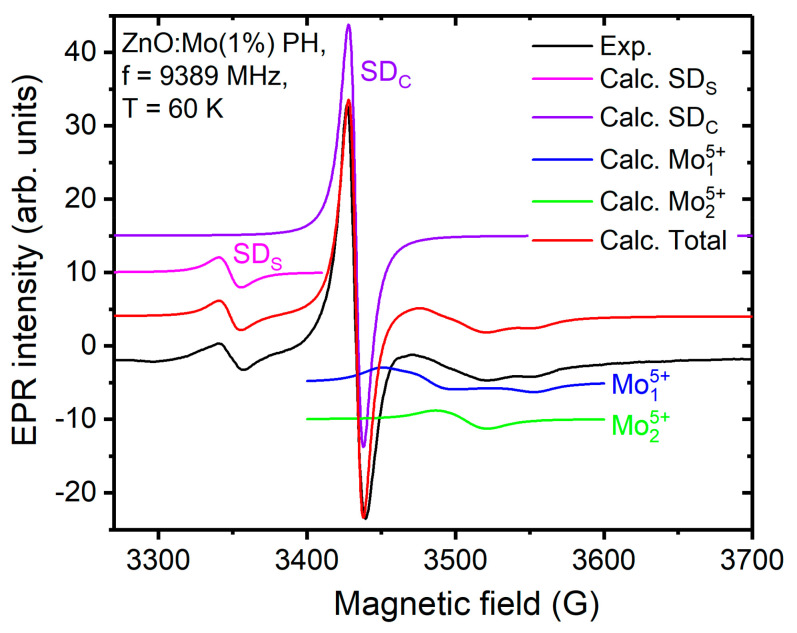
Experimental EPR spectrum measured in the ZnO:Mo(1%) PH sample. The signals calculated to fit the experimentally observed ones are indicated in the figure and in the legend.

**Figure 5 materials-15-02261-f005:**
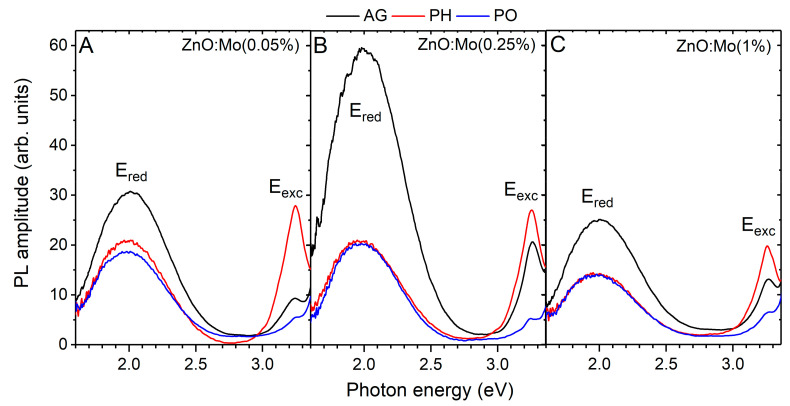
PL spectra measured at room temperature in: (**A**)—ZnO:Mo(0.05%), (**B**)—ZnO:Mo(0.25%), (**C**)—ZnO:Mo(1%) samples, as-grown and exposed to hydrogen or oxygen plasma.

**Figure 6 materials-15-02261-f006:**
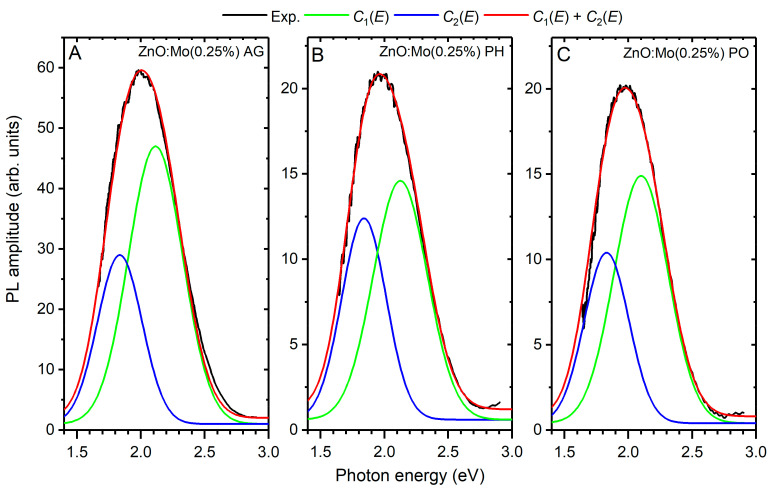
The decomposition of the experimental PL spectrum detected in the AG (**A**), PH (**B**) and PO (**C**) ZnO:Mo(0.25%) sample into the calculated Gaussian $C_{1,2}\left(E\right)$ components (as indicated in legends).

**Figure 7 materials-15-02261-f007:**
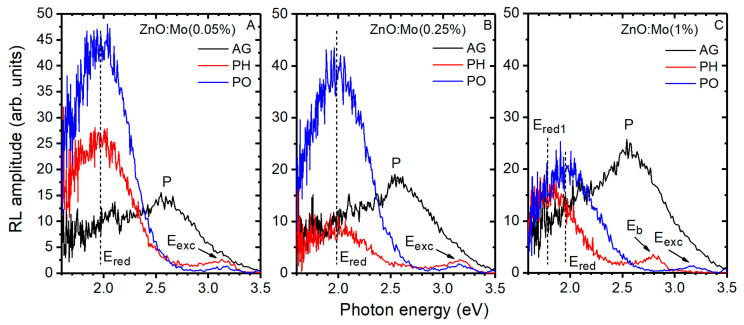
Room temperature RL spectra of ZnO:Mo(0.05%) (**A**), ZnO:Mo(0.25%) (**B**) and ZnO:Mo(1%) (**C**), AG, PH or PO. E_red_, E_red1_, E_b_, E_exc_ and P stress emission bands of different origin.

**Figure 8 materials-15-02261-f008:**
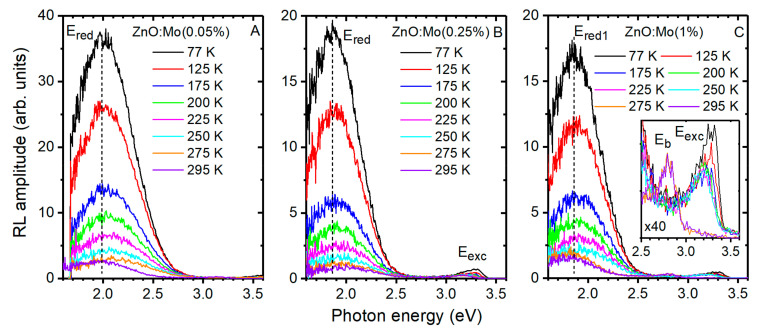
RL spectra of ZnO:Mo(0.05%) (**A**), ZnO:Mo(0.25%) (**B**) and ZnO:Mo(1%) (the inset demonstrates the closeup of the E_b_ and E_exc_ bands) (**C**) PH samples. Dashed vertical line is an eye guide for the changes in the position of the defect-related band.

**Figure 9 materials-15-02261-f009:**
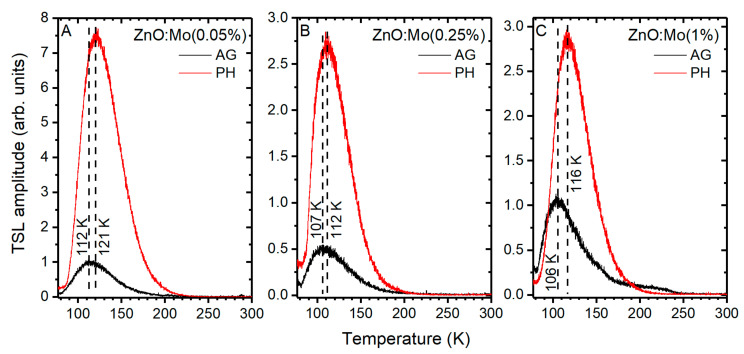
TSL glow curves of the ZnO:Mo(0.05%) (**A**), ZnO:Mo(0.25%) (**B**) and ZnO:Mo(1%) (**C**), AG and PH samples. Dashed vertical line segments are the eye guides to follow the glow peak positions.

**Figure 10 materials-15-02261-f010:**
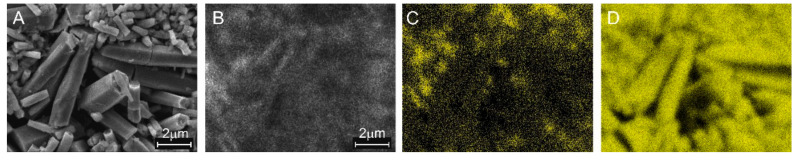
Secondary electron (SE) image (**A**), CL image (**B**), EDX signal for Mo (**C**) and EDX signal for zinc (**D**).

**Table 1 materials-15-02261-t001:** EPR integral intensity of the shallow donor signals.

Mo Content, %	SD_C_			SD_S_		
	EPR Integral Intensity, Arb. Units					
	AG	PH	PO	AG	PH	PO
0.05	15408	13077	14384	-	12	192
0.25	11768	12720	14219	-	344	1387
1	5995	8142	8716	-	1072	2688

**Table 2 materials-15-02261-t002:** $E_{\max1,2}$ and $G_{1,2}$ of the Gaussian components ($C_{1,2}(E)$) for each kind of treatment. The FWHM were $FWHM_{1}$ = 0.41 ± 0.01 eV and $FWHM_{2}$ = 0.5 ± 0.01 eV for all the $C_{1,2}(E)$ bands observed in all the samples.

**Nom. Mo Cont., %**	**T_ann_, °C**	$C_{1}(E)$		$C_{2}(E)$		Intensity Ratio $I(C_{1}(E)$$)/I(C_{2}(E))$
		$E_{\max1},\;\pm 0.01\;\mathrm{eV}$	$G_{1},\;\pm 5\;\mathrm{Arb}.\;\mathrm{Units}$	$E_{\max2},\;\pm 0.01\;\mathrm{eV}$	$G_{2},\;\pm 5\;\mathrm{Arb}.\;\mathrm{Units}$	
0.05	AG	1.84	135	2.12	230	2.66
	PH	1.84	118	2.13	153	2.03
	PO	1.82	85	2.10	135	2.48
0.25	AG	1.84	280	2.12	460	2.57
	PH	1.84	118	2.13	140	1.85
	PO	1.83	100	2.10	145	2.27
1	AG	1.84	108	2.12	178	2.58
	PH	1.84	72	2.13	90	1.95
	PO	1.83	64	2.10	92	2.25

## Data Availability

Not applicable.

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
