# Peer review of "Free-Standing ZnO:Mo Nanorods Exposed to Hydrogen or Oxygen Plasma: Influence on the Intrinsic and Extrinsic Defect States"

_materials, 2022, doi:10.3390/ma15062261_

Round 1
Reviewer 1 Report
With just cross-checking, the simarility of this work is too high (36%), especially it has been got 18% with previous work (M. Buryi, Z. Remeš, V. Babin, A. Artemenko, S. Chertopalov, J. Mičová. "Cold plasma treatment of ZnO:Er nano- and microrods: The effect on luminescence and defects creation", Journal of Alloys and Compounds, 2022). Therefore, the referee could not recommend publishing.
Author Response
The present manuscript text has been carefully compared with the mentioned previous work and all repetitions have been removed. Figures placed in the current submission contain data on the ZnO:Mo samples which on the first glance might have the view similar to the figures in the previous work but they are of absolutely different physical sense (different material) and, therefore, no copyright is needed.
Reviewer 2 Report
The paper entitled, ‘The influence of cold plasma treatment on molybdenum charge state, shallow donors and luminescence in ZnO:Mo nanorods’, by buryi et al is interesting. ZnO is an interesting material and its doping produces new properties that are important to understand from a fundamental point of view. The paper has a few concerns enumerated below.
- Line 71, ‘located in the spectral range about 1÷2 eV’ needs to be corrected
- The XRD of 0.05% Mo doping clearly shows the presence of Mo secondary phase. Please discuss phase separation and immiscibility of Mo. However, the secondary phase seemed to reduce with the treatments for this fraction of Mo. Please comment.
- There is a problem with caption of figures 2 and 3.
- In line 290, ‘…plasma treatment are plotted in Fig. 5.’, there is no figure 5.
- The PL part needs better description. There are several observations made on intensities of signals but the mechanism behind suppression of defects is not provided. For ex., on line 309, ‘Definitely, Mo doping level affects zinc vacancies (their charge state VZn0 = VZn2- + 2h+) responsible for the Ered band as discussed in 34,35’ is not sufficient. The mechanism should be clearly formulated for this case.
- Explain how charge compensation mechanisms due to Mo oxidation states affect the PL spectra.
- In figure 6, (D) figure is mentioned in the caption but is missing.
Author Response
R2: The paper entitled, ‘The influence of cold plasma treatment on molybdenum charge state, shallow donors and luminescence in ZnO:Mo nanorods’, by Buryi et al is interesting. ZnO is an interesting material and its doping produces new properties that are important to understand from a fundamental point of view. The paper has a few concerns enumerated below.
- Line 71, ‘located in the spectral range about 1÷2 eV’ needs to be corrected
A: The sentence has been corrected.
- The XRD of 0.05% Mo doping clearly shows the presence of Mo secondary phase. Please discuss phase separation and immiscibility of Mo. However, the secondary phase seemed to reduce with the treatments for this fraction of Mo. Please comment.
Thank you for noticing this. The corresponding comments have been added to the text (see first two paragraphs on p. 6)
- There is a problem with caption of figures 2 and 3.
A: The mentioned captions have been corrected.
- In line 290, ‘…plasma treatment are plotted in Fig. 5.’, there is no figure 5.
A: This is corrected now.
- The PL part needs better description. There are several observations made on intensities of signals but the mechanism behind suppression of defects is not provided. For ex., on line 309, ‘Definitely, Mo doping level affects zinc vacancies (their charge state VZn0 = VZn2- + 2h+) responsible for the Ered band as discussed in 34,35’ is not sufficient. The mechanism should be clearly formulated for this case.
A: The corresponding considerations and explanations have been added to the text on p. 10 below Figure 5 and on pp. 12, 13.
- Explain how charge compensation mechanisms due to Mo oxidation states affect the PL spectra.
A: The explanation has been added to the text on pp. 12,13.
- In figure 6, (D) figure is mentioned in the caption but is missing.
A: This has been corrected.

Reviewer 3 Report
The authors shows the influence of cold plasma treatment on ZnO:Mo nanorods via studying the molybdenum charge state, shallow donors and luminescence characteristics. The manuscript needs serious revisions before publication. I would like to decide based on the response to the following comments.
- The title is somewhat confusing. It would be better to revise the title and make it better understandable.
- Explain the main findings of the previous papers in literature 15 and 34 and explain your new contribution to the field in a whole new paragraph in the introduction section.
- Provide the exact amount in grams and the exact molarity of the chemicals used for doping in the experimental section. The section is explain is a way that the recipe is repeatable for the readers. The present explanation is incomplete.
- Provide the CAS numbers of all the chemicals used in the experiments.
- All the XRD peaks' orientation should be labled for clarity. Also, distinguish between the ZnO peaks and the peaks because of Mo (if any).
- Why the authors only considered the laid down nanorods for the application? I also recommend the authors to explain the formation chemistry of the nanorods provided the morphology.
- It is claimed that the nanorods are perfectly hexagonal. However, I can see that most of the nanorods are not perfectly hexagonal. Explain the reason in the manuscript.
- I recommend the authors to perform EDS and XPS to confirm if the dopant was successfully doped into ZnO or not?
- I recommend the authors to perform TEM to confirm the defect states after hydrogen and oxygen plasma treatments of the doped samples.
Author Response
R1: The authors shows the influence of cold plasma treatment on ZnO:Mo nanorods via studying the molybdenum charge state, shallow donors and luminescence characteristics. The manuscript needs serious revisions before publication. I would like to decide based on the response to the following comments.
The title is somewhat confusing. It would be better to revise the title and make it better understandable.
A: Thank you for this remark. The title has been changed: “Free-standing ZnO:Mo nanorods exposed to hydrogen or oxygen plasma: Influence on intrinsic and extrinsic defect states”.
R1: Explain the main findings of the previous papers in literature 15 and 34 and explain your new contribution to the field in a whole new paragraph in the introduction section.
The new paragraph has been added at the end of Introduction.
R1: Provide the exact amount in grams and the exact molarity of the chemicals used for doping in the experimental section. The section is explaining a way that the recipe is repeatable for the readers. The present explanation is incomplete. Provide the CAS numbers of all the chemicals used in the experiments.
A: It has been done. The following text was included in the manuscript:
“All the chemicals were used as-received without additional purification. Zinc nitrate hexahydrate (CAS: 10196-18-6) (Zn(NO3)2⋅6H2O) and hexamethylenetetramine (CAS: 100-97-0) (HMTA, C6H12N4) were purchased from Slavus. As Mo dopant precursor was used ammonium heptamolybdate tetrahydrate (CAS: 12054-85-2) (NH4)6Mo7O24⋅4H2O) (NHMO) from Sigma-Aldrich. Deionized water was purified with a system So ~ Safe Water Technologies, having a conductivity 0.20 μS⋅cm-1 at 25°C. Mo-doped ZnO nanorods with nominal compositions Zn1-xMoxO (x = 0, 0.0005, 0.0025, 0.01) were prepared by the hydrothermal growth method. First, the Zn(NO3)2⋅6H2O and (NH4)6Mo7O24⋅4H2O in the corresponding molar ratio (99.95:0.05 or 99.75:0.25 or 99:1) were dissolved in 250 ml of deionized water, with vigorous stirring by a magnetic stirrer at RT for 30 min. The sample Zn0.9995Mo0.0005O was prepared from 1.8583 g (24.9876 mM) of Zn(NO3)2⋅6H2O and 0.0039g ( 0.0124 mM) (NH4)6Mo7O24⋅4H2O. 1.8545 g (24,9376 mM) of Zn(NO3)2⋅6H2O and 0.0193 g (0.0624 mM) of (NH4)6Mo7O24⋅4H2O were used to prepare sample Zn0.9975Mo0.0025O. The sample Zn99MoO was prepared from 1.8406 g (24,75 mM) of Zn(NO3)2⋅6H2O and 0.0772 g ( 0.25 mM) of (NH4)6Mo7O24⋅4H2O. The nominal concentration of the Zn1-xMoxO (x = 0, 0.0005, 0.0025, 0.01) in the final suspensions was 25 mM. Second, was prepared aqueous solution of HMTA 0.8762 g (25 mM) (250 ml) (the same amount for the preparation of all samples). All the solutions were filtered through a Whatman 2 filter. Subsequently, the HMTA solution was added to each Zn1-xMoxO solution and were stirred vigorously at RT for 15 min. Third, the reaction mixture was followed by hydrothermal growth of nanorods by heating at 90◦ C for 3 h. The grown nanorods were isolated and then purified (removal of residual salts), washing 5 times with deionized water and subsequent centrifugation at 11,600 rpm (RCF: 15,500 × g) for 15 min. Finally, the samples were lyophilized at least 24 hours”.
R1: All the XRD peaks' orientation should be labelled for clarity. Also, distinguish between the ZnO peaks and the peaks because of Mo (if any).
A: It has been done. For better clarity two new figures have been added to Supplementary Information (see Figs. S1,2).
R1: Why the authors only considered the laid down nanorods for the application? I also recommend the authors to explain the formation chemistry of the nanorods provided the morphology.
Indeed, we considered both free-standing nanorods (laid down) and deposited on the substrate. The present manuscript is dedicated to the free-standing ones. The second manuscript which is currently under preparation deals with the deposited nanorods and will be published elsewhere. The free-standing nanorods are much cheaper in the preparation, since there is no need in the expensive substrate and nucleation layer to be deposited onto it. As for the formation chemistry – it can be explained as follows (also included in the text on p. 6 below Figure 2). The nanorods are grown from a water solution containing dissolved precursors. The solution contains chaotic nucleation centers – seeds, very small pieces (at maximum tens of nm large) of not fully dissolved reagents. The solution is kept at the temperature (maintained during the whole growth process) specific for the nucleation to begin. This temperature has been found experimentally (see Refs. 1-3,6) after numerous attempts to grow namely rod-like structures, i.e., the geometry of the rods is the product of the tiny balance between the expansion due to growing of the ZnO crystalline and dissolution in the solvent. The corresponding calculations were not carried out.
R1: It is claimed that the nanorods are perfectly hexagonal. However, I can see that most of the nanorods are not perfectly hexagonal. Explain the reason in the manuscript.
A: Oppositely, the majority of the structures is hexagonal rods. To demonstrate this, the images with better contrast have been taken. They are shown in Figure 2 instead of the previous ones. Moreover, for better clarity three new figures have been added to Supplementary Information (see Figs. S3-5).
R1: I recommend the authors to perform EDS and XPS to confirm if the dopant was successfully doped into ZnO or not?
As for the XPS analysis – it was performed on the same set of samples as the one presently used. The results are published in another paper (see Ref. 33 in the list of references). There is also complete analysis of the X-ray induced Mo5+ ions detected by EPR. Combination of these two techniques was enough to confirm the presence of the Mo dopant in the ZnO host in Ref. 33. Moreover, the same set of samples has been annealed in air and studied by EPR in another paper (see Ref. 20). And, again, the Mo5+ signals were confirmed to originate from the ZnO host there. Therefore, based on these two previous works, one may conclude that Mo indeed exists in the ZnO structure. These considerations have been added to the text. EDS has been performed and the results are presented in the revised version of the manuscript as well. They also confirm the presence of Mo.
R1: I recommend the authors to perform TEM to confirm the defect states after hydrogen and oxygen plasma treatments of the doped samples.
TEM experiments were indeed carried out on the presently considered set of samples. However, the obtained results are the part of the new paper which will be published elsewhere.

Round 2
Reviewer 1 Report
it is OK now
Author Response
No COPYRIGHT is needed since all figures in the current submission are original. Seeming resemblance between the current submission and the previous papers originates from the fact that similar bunch of experimental methods has been used.
Reviewer 2 Report
The paper can now be accepted for publication as the authors have replied to all the comments of this reviewer.
Author Response

(The authors gave the same response as above.)

Reviewer 3 Report
The authors have provided a detailed response to the review comments. Most of the responses are satisfied and the revised manuscript looks in good shape. I believe the manuscript is now ready for publication. However, I want the authors to consider the following point to further improve the research idea and the manuscript content for readers' understanding.
Comment: I would be great if the authors could provide a brief comparison of the effects of plasma treatment on free standing and deposited on surface nanorods. The authors can either use their own results or use a reference paper to present the comparison.
Good Luck!
Author Response
We are grateful to the reviewer for the valuable comments including the current one. We have taken it fully into account and added a small paragraph with the comparison of the present results and the results available from the world literature (for the deposited ZnO nanorods) on p. 12 just at the end of the subsection 3.3 (lines 459-474). Three new references have been added as well.